# An In Vivo Spectrophotometric Analysis of Gingival Acrylic Shade Guide

**DOI:** 10.3390/ma14071768

**Published:** 2021-04-03

**Authors:** Peter C. Grieco, John D. Da Silva, Yoshiki Ishida, Shigemi Ishikawa-Nagai

**Affiliations:** 1Department of Restorative Dentistry and Biomaterials Science, Harvard School of Dental Medicine, 188 Longwood Avenue, Boston, MA 02115, USA; peter_grieco@hsdm.harvard.edu; 2Department of Oral Medicine, Infection, and Immunity, Harvard School of Dental Medicine, 188 Longwood Avenue, Boston, MA 02115, USA; yoshiki_ishida@hsdm.harvard.edu

**Keywords:** color of gingival tissue, color of gingival acrylic material, spectrometric analysis, spectrophotometer, CIELAB, shade guide

## Abstract

Selecting shades of acrylic gingival restorative material is challenging. This study examined the shade appropriateness of five acrylic gingival restorative materials. The color was analyzed using an intraoral spectrophotometer (Crystaleye^®^, Olympus). The gingival color of maxillary incisors for eighty-nine patients was measured. CIELAB color coordinates (L*, a* and b*) were obtained, and the color difference ∆E (Coverage Error: CE) between shade tabs and natural gingival color of patient samples for each shade guide system were compared. Repeated ANOVA and post hoc analyses with Tukey′s HSD were performed. There was a significant difference among the mean minimum CEs of the tab sets (*p* < 0.01). GC Acrylic (CE = 5.89 ∆E ± 2.97) and Lucitone 199^®^ (CE = 6.55 ± 3.33) groups exhibited CEs significantly lower than all other groups (all *p* < 0.001). The IvoCap^®^ system exhibited the highest CE (10.78 ± 3.80), significantly greater than all other groups (*p* < 0.001). No significant differences were observed based on sex (*p* = 0.055) or ethnicity (*p* = 0.327). The GC Acrylic and Lucitone 199^®^ shade guides showed the lowest CEs. All guides had coverage errors above 5.89 ∆E, which is larger than ∆E thresholds of acceptability. Of the materials evaluated in this study, GC Acrylic and Lucitione 199^®^ are best able to reproduce the clinical appearance of the gingival tissue. Many patients have tissue that cannot be reproduced accurately with currently available materials.

## 1. Introduction

Although great progress has been made in reproducing the esthetics of dental hard tissues, progress in the prosthetic reproduction of gingival tissues lags [1]. This is a critical area of focus because the overall acceptance of, and patient satisfaction with, a prosthesis is dependent on the esthetics of both gingival tissue and teeth [2,3,4,5]. When replacing lost gingival tissue in the esthetic zone, the morphology and appearance of the prosthetic gingival tissue is essential for an optimal esthetic outcome [6,7,8,9]. To meet these esthetic demands, many of the same factors that are taken into account when replacing teeth, such as gingival color, translucency, texture, and inflammatory status. These factors must be considered, interpreted, documented, and then, ideally communicated to the technician for prosthetic replication. Further, an objective means for assessing the spectral properties of gingival tissues and restorative materials is necessary both to acquire and communicate these gingival reflectance attributes and to determine the current appropriateness and areas of need for gingival restorative materials [10]. However, scant research exists on the threshold of perceivable difference in gingival color to an observer. Classic research into the perceptibility of color differences in the oral environment originally defined threshold level of perceptibility as less than 3.7 ∆E units in the CIELAB color space [11,12,13,14,15]. Subsequently, it has been established that the spatial color difference of 1 ∆E unit can be perceived by approximately 50% of experienced observers [16,17]. Further, studies have established different levels of perceptibility for differences in varying prosthetic applications. Levels of difference required for discernment have ranged from ∆E = 2.6 for denture teeth [18] to ∆E = 1.6 for all-ceramic crowns [16,19]. Currently, little research exists delineating the ∆E at which gingival tissues can be discerned [1,10]. One study, performed by Paniz et al., attempted to establish the threshold at which gingiva of differing optical properties could be distinguished. In a study of 39 patients, this threshold was determined to be at ∆E = 8.74. These results suggest that there is a higher threshold of esthetic acceptability in gingival tissues than in dental structures. Material-specific problems also exist when using gingival composite materials, as the coverage ability of gingival materials has been shown to vary significantly depending on the incident light source [1].

Research into the clinical acceptability of currently available gingival shade guides is also scant. In a pilot study, Ito et al. explored the creation of a novel shade guide in a population using a cluster analysis [20], but the appropriateness of existing guides was not explored. Additionally, minimal in vivo research currently exists on the accuracy of gingival replacement materials in optically replicating natural gingival structures. This is likely due to the difficulty of intraoral soft tissue shade acquisition using a contact-type dental spectrophotometer. The purpose of this study was to examine and determine the appropriateness of five different commercially available acrylic gingival restorative materials in a sample patient population using CIELAB color coordinates ∆L*, ∆a* and ∆b*, and ∆E (Coverage error: CE) and its difference among sex and ethnic backgrounds.

## 2. Materials and Methods

This study was performed at Harvard School of Dental Medicine with the approval of its Institutional Review Board (IRB 14-1246). Patients were recruited from among the general patient population of the Harvard Dental Center. After recruitment, screening of eligibility as an active patient of the Dental Center, and patient agreement to participate, a baseline periodontal examination of the maxillary central incisors was performed. Inclusion criteria for the study were: Patient age eighteen and above; either tooth #8 or #9 present with no restorations, a vital directly restored tooth, or a vital tooth with an all-ceramic or ceramo metal full-coverage restoration. Exclusion criteria were: ADA class 3 or greater, bleeding on probing to buccal sites of teeth #′s 8 or 9, or probing depths greater than 3 mm in teeth #′s 8 or 9. Instances of racial pigmentation were not excluded.

### 2.1. Color Measurement of Attached Gingiva

For each participant, an area of the attached gingiva 3 mm wide by 1 mm high located 3 mm apical to the free gingival margin of a single anterior tooth was evaluated. Tooth #8 was primarily chosen for analysis. If tooth #8 did not fit the inclusion criteria, tooth #9 was used with the identical protocol. Using an intraoral dental spectrophotometer (Crystaleye; Olympus, Tokyo, Japan), spectral [21] images of the study area were obtained, ensuring that no pressure was placed upon the area and that the area was isolated from ambient light sources (Figure 1). Reflectance values were transferred from the spectrophotometer to a personal computer and CIELAB color coordinates L*, a*, and b* were obtained.

### 2.2. Color Measurement of Gingival Shade Guides

In vitro analysis was then performed using like conditions on the gingival restorative materials. Five different commercially available PMMA gingival restorative material systems were selected. Systems chosen for analysis included: GC Acrylic (GRADIA Gum Shades^®^; GC America, Inc., Alsip, IL, USA), IvoCap^®^ (Ivoclar Vivadent, Inc., Laguna, Philippines), Lucitone 199^®^ (Dentslpy International, Hong Kong, China), Easy-Flow^®^ (Henry Schein, Hong Kong, China), and Candulor^®^ (Ivoclar Vivadent, Inc., Laguna, Philippines). These systems represented the most popular gingival restorative materials from a survey of local and regional dental laboratories.

Current, complete commercial shades tab sets for each system were obtained. Shade tabs were adjusted by removing from the posterior surfaces such that the registered area of each sample was resultantly uniformly 1 mm thick and the surface was left glossy. Front-facing shade tab surfaces were not adjusted from states from which they were distributed by manufacturer. For each system, all of the constituent shade tabs were individually placed into a specially designed isolation box and specimen holder (Crystaleye^®^ laboratory inspection box; Olympus), such that the tab orientation and distance from the spectrophotometer lens was identical to the intraoral environment. Three readings 3 mm in width by 1 mm in height were obtained from center area of each shade tab (Figure 2). Reflectance values were transferred from the spectrophotometer to a personal computer for analysis using CIELAB color coordinates. The mean CIELAB values for each of the shade tabs, comprising five shade guide sets were calculated.

### 2.3. Color Comparison between Natural Gingiva and Shade Guides

The color difference between each of natural gingiva and shade tab were calculated in ∆L*, ∆a*, ∆b*, and ∆E by following Equations (1)–(4).
∆L*: L*_(shade tab)_ − L*_(natural gingiva)_(1)
∆a*: a*_(shade tab)_ − a*_(natural gingiva)_(2)
∆b*: b*_(shade tab)_ − b*_(natural gingiva)_(3)
∆E: (∆L* ^2^ + ∆a* ^2^ + ∆b* ^2^)^1/2^(4)

The Coverage Error (CE) was then determined. CE is the average of ΔE value between a patient’s shade and the best match that each shade guide system. For each of the 89 sites, the shade tab from each system with the smallest ΔE was identified and was denoted the “best match tag” for that system. The CE for a shade guide system was, therefore, the average ΔE between each of the 89 sites and the best match shade tab with the minimum ΔE to that site. The ∆L*, ∆a*, and ∆b* of the best matched shade tab were also analyzed by CIELAB color coordinates distribution graph.

### 2.4. Statistical Analysis

The CE, ∆L*, ∆a*, and ∆b* of each shade guide system were then compared. Using statistical modeling software (SPSS; IBM, Armonk, NY, USA), one-way ANOVA analyses were performed to determine if statistically significant differences existed in coverage area among the five different shade guides. Thereafter, if necessary, a post hoc analysis was performed using Tukey HSD (α = 0.05). The CEs of the differing shade guides were then stratified with respect to participant sex and ethnicity. An independent t-test (α = 0.05) was performed to determine if statistically significant differences existed between mean coverage error and participant sex. One way ANOVA analyses (α = 0.05) were performed to determine if statistically significant differences existed between mean coverage error and participant ethnicity. Thereafter, if necessary, a post hoc analysis was performed using Tukey’s HSD (α = 0.05).

## 3. Results

Eighty-nine individuals participated in this study. Tooth #8 was used in seventy-nine analyses and, due to implants present in the #8 site, tooth #9 was used in ten analyses. Participants were asked to self-identify for age, sex, and ethnicity. Patient ages ranged from 25 to 87 with a median of 39 years. There were 48 females and 41 males. Forty-two participants identified as White/Caucasian, twenty as Asian, fourteen as Black/African American, and thirteen as Hispanic.

The mean values of overall CE for each shade guide system are shown in Table 1. The ANOVA analyses indicated significant differences in the overall CE between the groups of individual shade guide sets. Post hoc analyses determined that the IvoCap^®^ CE was significantly higher than that of all other materials (*p* < 0.001 for all comparisons). The GC Acrylic CE was significantly lower than the IvoCap^®^ (*p* < 0.001), Easy-Flow^®^ (*p* < 0.001), and Candulor^®^ (*p* < 0.001) groups. The Lucitone 199^®^ CE was significantly lower than the IvoCap^®^ (*p* < 0.001), Easy-Flow^®^ (*p* < 0.001), and Candulor^®^ (*p* < 0.001) groups. The GC Acrylic and Lucitone 199 groups exhibited the lowest CEs and exhibited no statistically significant difference from each other. Color difference coordinators (ΔL*, Δa*, Δb*) between the best match shade tab and patient′s shade were plotted in Figure 3 with zero representing the patient’s shade. There are 2 characteristics; direction and extent, of how shade tab′s color differs from the patient′s shade. The data points of ∆a* − ∆b*, in quadrant I (Figure 3b–f) on the top right indicated that the best matched shade tab presented with redder (less green) and yellower (less blue) compared to the color of natural gingiva for that participant. Data points in quadrant II on the bottom right represents shade tabs with redder (less green) but less yellow (bluer). Data points in quadrant III on the bottom left represents tabs with less red (greener) and less yellow (bluer). Data points in quadrant IV on the top left represents tabs with less red (greener) and more yellow (less blue). For the ΔL*, the plot distribution (Figure 3a) explains the direction of color shifts and extent of the best matched shade tabs compared to the natural gingiva. The GC Acrylic and Lucitone 199^®^ indicated a smaller and closer zero point plot distributions on ∆a* − ∆b* map (Figure 3b,c). In contrast, the other 3 systems indicated wider distributions towards quadrant II, especially IvoCap^®^, which indicated a larger shift to quadrant II (Figure 3d–f). For ΔL*, GC Acrylic indicated the smallest distribution away from the zero point (Figure 3a), while other 4 systems indicated distribution toward lower ΔL* (darker). Statistical analysis indicated that GC Acrylic showed significantly smaller ΔL*and Δb* than theother 4 systems (*p* < 0.01, Figure 3a,c), and both GC Acrylic and Lucitone 199^®^ indicated smaller Δa* than Ivo-Cap^®^ and Candulor^®^ (*p* < 0.01, Figure 3b).

When stratified according to patient sex, the female group best matching acrylic material exhibited a mean CE of ∆E = 8.15 ± 4.10, whereas the male group exhibited ∆E = 7.46 ± 3.56. The independent sample t-test found this difference not to be statistically significant (*p* = 0.062). The mean ∆E between each participant and the best match of each shade guide set was then stratified with respect to participant ethnicity. When stratified according to patient ethnicity, the white group best matching acrylic material exhibited a mean CE of ∆E = 7.60 ± 3.85, the Hispanic group exhibited a mean CE of ∆E = 7.43 ± 4.23, the black group a mean CE of ∆E = 8.12 ± 4.33, and the Asian group a mean CE of ∆E = 8.44 ± 3.91. There was no statistical difference among the ethnicity groups (ANOVA, *p* = 0.327).

Lastly, the percentage of participants who were ‘matched’ with a clinically unacceptable shade was calculated and examined by tabulating the number of participants above the ΔE threshold proposed by Paniz 2014, Ren 2015, Sailer 2014, and Paravina 2018 for each shade guide, as a fraction of the total patient pool. The percentage of patients with best match above this threshold of clinical acceptability is shown in Table 2. Based on the threshold reported by Paniz, 81% of participants had clinically acceptable shade tab with GC Acrylic, 76.7% with Lucitone 199^®^, 62.2% with Easy-Flow^®^, 54.4% with Candulor^®^, and only 31.2% with Ivo-Cap^®^. Using Sailer′s threshold, the percentage of acceptable shade tabs were 15.7%, 12.4%, 0%, 0.2%, and 10.1%, respectively. Although this trend was observed with other 2 thresholds reported by Ren and Paravina.

## 4. Discussion

For the overall population, the mean CE for all materials varied from a mean of 5.89 for GC Acrylic to a mean of 10.78 for IvoCap^®^. The clinical significance of these differences is dependent on the threshold of acceptability that is used in the analysis. There are several studies reported that discuss the threshold of clinical acceptability in the intraoral gingival areas. Paniz et al. state that a clinically acceptable shade match in gingival is ΔE = 8.74 [1], while Sailer et al. propose that the match be much more precise for acceptability, ΔE = 3.1 [22]. Ren et al. in 2015, and Perez and Paravina et al. in 2018, proposed 50:50 acceptability thresholds, with their threshold levels being calculated as ΔE = 3.7 [23,24] and ΔE = 4.0 [25], respectively. Table 2 illustrates the percentage of participants in this study, that when using each of the different acceptability thresholds, would be left with an unacceptable color match in each shade system. While the results differ based on the acceptability threshold used, in a material such as the IvoCap^®^, the mean best match (ΔE = 10.78) is outside the level of clinical acceptability of any of the thresholds. For that shade system, depending on threshold used, 67.8% to 98.9% of patients will have best matches above the threshold of clinical acceptability, which is a clinically significant problem.

Via analysis of the ΔL* and individual Δa* − Δb* charts, the means in which each shade guide′s match varies from the patient population can be examined. For all sets, the ΔL* values of the best shade match were lower than that of the patient population, indicating that the majority of shade tabs analyzed are darker than the patient gingiva they replace. Similarly, analysis of the Δa* − Δb* values of the best matches reveals that the majority of the best matches reside in the +Δa* range. This indicates that the majority of the gingival restorative materials are clustered in the lower right quadrant. This area indicates that the best match shade tabs are chromatically bluer and redder than the patient gingiva they are replacing. When these two values are interpreted together, the best match shade tabs can be generalized as being too purple. Shade guides clustered more towards the center of each of the ΔL* and Δa* − Δb* charts are also those with the lowest coverage errors. This is best indicated by the GC acrylic, which exhibits a value difference (ΔL*) that is centered about ΔL = 0, while its chroma (Δa* − Δb*) chart is clustered the closest to the center of the graph. These two charts indicate a best match of both value and chroma values for this material, and this accuracy is reflected in the low coverage area for this material.

There are also limitations involved in the study′s analysis. A uniform 1 mm thickness of materials was used for all material data acquisitions. In patient care, thicknesses greater or less than this amount may be used, effecting the translucency and color-matching ability of the material. Further, the standardization method for measurement may have an effect on the measured L* value of the material, when compared to the natural gingival shade. The measurement of material was performed using a free-standing mount with colorless black background, and shade tabs uniformly 1 mm thick to allow passage of the illumination light of the spectrophotometer for measurement. This black background may result in a lower observed L* value.

Further, only a limited selection of acrylic types of shade tabs were utilized for this study. Materials that may be used clinically to replicate gingival tissue also include pink composite and pink porcelain materials. Future research would be indicated to evaluate the coverage error of shade guides used for other types of materials. Another complicating factor in evaluating gingival color is the propensity for gingival color to fluctuate, even in the same patient, according to several factors. These factors include alterations in the gingival blood flow as well as the presence or absence of gingival inflammation. With gingival shade being variable, it is possible for the coverage error to fluctuate as the shade changes towards or further away from that of the shade tab set. This effect can be minimally controlled by attempting to match gingival shades in healthy sites free of periodontal disease and bleeding on probing.

With regard to participant sex and ethnicity, our research suggests that there is not enough variation in the respective gingival optical properties of males and females or amongst different ethnic groups to suggest a significant difference in the coverage error of the studied selection of gingival restorative materials based on sex and ethnicity. Another factor to be considered is age. Although our study population age varies 25~87, further investigation of the different age groups would be desirable.

Future studies are needed, however, with increased sample size and across more varying populations, to validate the above findings. However, if accurate, this study may provide a compelling basis for altering existing shade coverage or adding new shades to existing sets to help decrease the coverage error in gingival restorative materials.

## 5. Conclusions

Within the limitations of this study, the following conclusions were drawn:(1)The GC Acrylic and Lucitone 199^®^ shade guides showed the lowest CEs compared to Easy-flow^®^, IvoCap^®^, and Candulor^®^ systems.(2)All shade guides exhibited coverage errors above 5.89 ∆E which is larger than ∆E thresholds of acceptability.

Well color matched gingival acrylic materials are desired to establish better esthetic outcomes. The gingival tissue color for many patients cannot be matched reliably.

## Figures and Tables

**Figure 1 materials-14-01768-f001:**
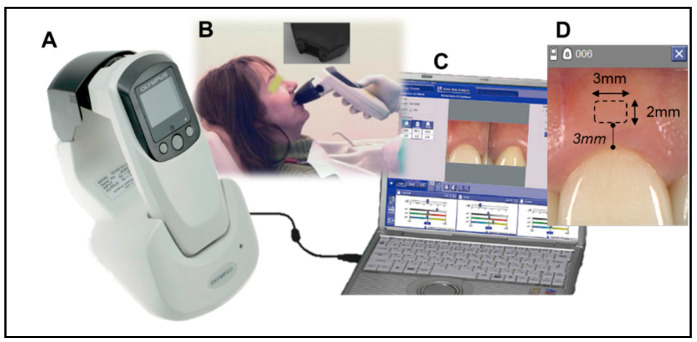
Spectrophotometric color measurement of the gingiva. (**A**): Spectrophotometer Crystaleye^®^. (**B**): Measurement is taken using a special rubber-cap for non-contact measurement. (**C**): The spectrophotometric data is transferred to the Crystaleye^®^ software for color data analysis. (**D**): The color reading of the rectangular area (3 mm × 2 mm) 1 mm above the tooth was obtained.

**Figure 2 materials-14-01768-f002:**
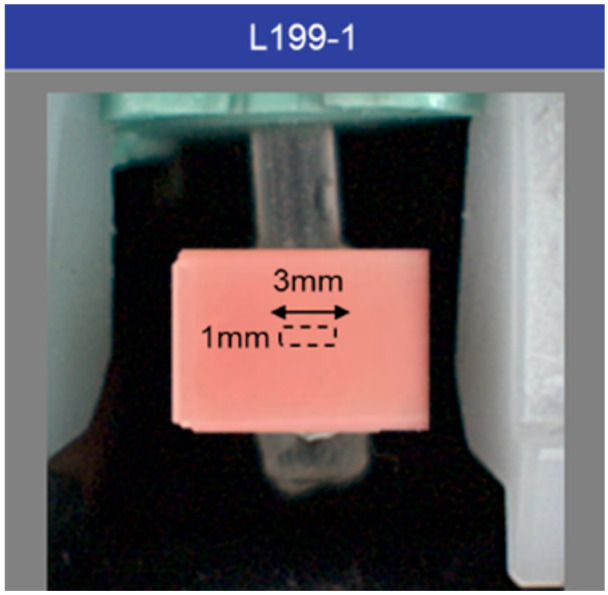
Spectrophotometric color measurement of gingival shade guides. The color reading of the rectangular area (3 mm × 1 mm) in the center of the shade guide tab was obtained.

**Figure 3 materials-14-01768-f003:**
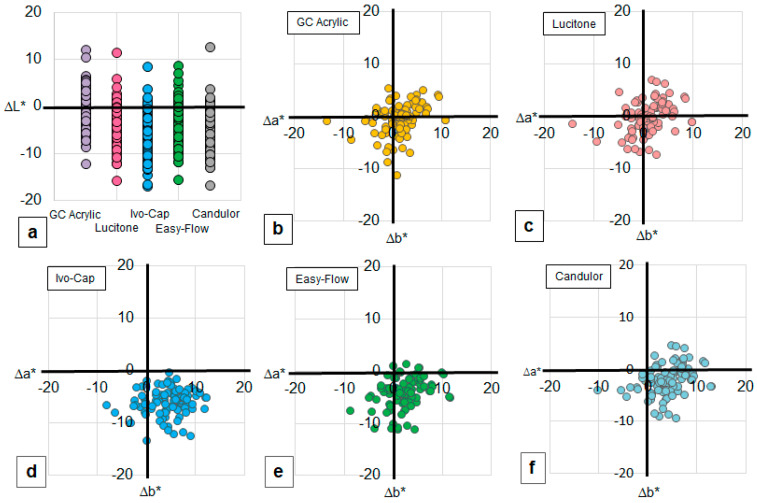
Direction and extent of CIELAB color coordinates with zero representing the patient′s shade. (**a**) ∆L* distribution, (**b**) ∆a* − ∆b* distribution of GC acrylic, (**c**) Lucitone, (**d**) Ivo-Cap, (**e**) Easy-Flow, (**f**) Candulor.

**Table 1 materials-14-01768-t001:** Coverage Error (CE) for each shade guide system and gingival color of study participants (Mean and SD).

∆E (CE)	5.89 (2.97)	6.55 (3.33)	10.78 (3.80)	7.92 (3.54)	8.13 (3.85)
	GC Acrylic	Lucitone 199^®^	IvoCap^®^	Easy-Flow^®^	Candulor^®^
GC Acrylic		NS	(*p* = 0.001)	(*p* = 0.001)	(*p* = 0.001)
Lucitone 199^®^			(*p* = 0.001)	NS	(*p* = 0.025)
IvoCap^®^				(*p* = 0.001)	(*p* = 0.001)
Easy-Flow^®^					NS

**Table 2 materials-14-01768-t002:** The percentage of patients with best match shade tab above the acceptable threshold.

Color Threshold	Beyond Clinical Unacceptable Match ∆E 8.7 by Paniz 2014	Beyond 50:50% Acceptability Threshold ΔE 4.0 by Ren 2015	Beyond Mean ∆E Threshold 3.1 by Sailer 2014	50:50% Acceptability Threshold 3.7 by Perez and Paravina 2018
GC Acrylic	18.8%	74.4%	84.3%	81.1%
Lucitone 199	23.3%	76.5%	87.6%	81.1%
IvoCap	67.8%	98.9%	100%	98.9%
Easy-Flow	37.8%	85.6%	98.9%	92.2%
Candulor	45.6%	87.8%	89.9%	87.8%

## Data Availability

The data presented in this study are available on request from the corresponding author. The data are not publicly available due to institutional restrictions.

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
