# Peer review of "An In Vivo Spectrophotometric Analysis of Gingival Acrylic Shade Guide"

_materials, 2021, doi:10.3390/ma14071768_

Round 1
Reviewer 1 Report
This paper examed shade appropriateness of five acrylic gingival restorative materials by an in vivo spectrophotometric analysis.While the The results are very interesting, there are some problems need to be revised or explained:
- In the manuscript, the introduction for some abbreviations is missing at first mention, e.g. CE.
2.The color of human gingiva becomes lighter with age. The color differences between different age groups should to be concerned.
3.The conclusion should be properly detailed
Author Response
Comments and Suggestions for Authors
This paper examed shade appropriateness of five acrylic gingival restorative materials by an in vivo spectrophotometric analysis. While the The results are very interesting, there are some problems need to be revised or explained:
- In the manuscript, the introduction for some abbreviations is missing at first mention, e.g. CE.
Thank you for the comment.
We revised introduction and abstract.
Introduction
using CIELAB color coordinates ∆L*, ∆a* and ∆b*and ∆E (Coverage error: CE).
Abstract
CIELAB color coordinates (L*, a* and b*) were obtained and the color difference ∆E (Coverage Error: CE) between…
- The color of human gingiva becomes lighter with age. The color differences between different age groups should to be concerned.
Thank you for the suggestion.
Patient ages ranged from 25 to 87, but sample size was not big enough to analyze age groups. We included factor of “age” in the limitations as below.
“Another factor to be considered is age. Although our study population age varies 25 ~ 87, further investigation of the different age groups would be desirable”.
- The conclusion should be properly detailed
Thank you for the suggestion.
We revised conclusion adding summary as below.
2) All shade guides exhibited coverage errors above 5.89∆E that is a larger than ∆E thresholds of acceptability. “Well color matched gingival acrylic materials are desired to establish better esthetic outcomes. The gingival tissue color for many patients cannot be matched”.
Reviewer 2 Report
The authors covered an interesting topic, gingival acrylic is something that is often overlooked. The manuscript is well written, the study is simple and well designed. The sample size should be increased to detect significant differences. The authors gave a good review and explanation of the limitations of the study. I recommend publication of this manuscript.
Author Response
Comments and Suggestions for Authors
The authors covered an interesting topic, gingival acrylic is something that is often overlooked. The manuscript is well written, the study is simple and well designed. The sample size should be increased to detect significant differences. The authors gave a good review and explanation of the limitations of the study. I recommend publication of this manuscript.
Thank you very much.
Reviewer 3 Report
In this prospective in-vivo study, the authors determined the appropriateness of five different commercially available acrylic gingival restorative materials in 89 adults using CE and CIELAB color coordinates and its difference among sex and ethnic backgrounds. Therefor the differences between best-fit color of the restorative material and natural gingiva was calculated (coverage error). For measurement, a scientifically well-established intraoral dental spectrophotometer has been used.
English language is well. The keywords are appropriate. The statistics are simple, but sufficient. There are some minor points to address.
Abstract & conclusion: You report “All guides coverage errors above 5.89ΔE” What is the meaning of this threshold? Please do not present this threshold without clinical interpretation.
No ethical approval number is presented.
There are no information for validity and reliability of the spectrophotometer used presented.
Why did the authors perform-post-hoc t-tests for participant ethnicity, when the ANOVA already showed no significant difference for coverage error?
Figure 3 shows excessive multiple testing without statistical compensation. I recommend dropping figure 3 and include the statistical test values in table 1 for ΔE only.
Figure 5 The color schema for “above threshold” has no match within the graph and column 3 contains a typo (68 % instead of 67 %). Furthermore, figure 5 contains redundant information from table 2. I recommend removing figure 5.
Missing literature:
doi: 10.1016/j.jpor.2010.12.005
doi: 10.1016/j.jdent.2011.10.001
Author Response
Comments and Suggestions for Authors
In this prospective in-vivo study, the authors determined the appropriateness of five different commercially available acrylic gingival restorative materials in 89 adults using CE and CIELAB color coordinates and its difference among sex and ethnic backgrounds. Therefor the differences between best-fit color of the restorative material and natural gingiva was calculated (coverage error). For measurement, a scientifically well-established intraoral dental spectrophotometer has been used.
English language is well. The keywords are appropriate. The statistics are simple, but sufficient. There are some minor points to address.
- Abstract & conclusion: You report “All guides coverage errors above 5.89ΔE” What is the meaning of this threshold? Please do not present this threshold without clinical interpretation.
Thank you for the feedback.
We revised abstract and conclusion accordingly as below.
Abstract
All guides coverage errors above 5.89∆E which is larger than ∆E thresholds of acceptability.
Conclusion
All shade guides exhibited coverage errors above 5.89∆E which is larger than ∆E thresholds of acceptability.
- No ethical approval number is presented.
Thank you for the feedback.
Institutional Review Board (IRB) approval number IRB 14-1246 was added.
- There are no information for validity and reliability of the spectrophotometer used presented.
Thank you for the feedback.
This spectrophotometer was FDA approved and used for many publications. We added one additional reference “Dental color matching instruments and systems”
Chu SJ, Trushkowsky RD, Paravina RD. Dental color matching instruments and systems. Review of clinical and research aspects. J Dent. 2010;38 Suppl 2:e2-16.
- Why did the authors perform-post-hoc t-tests for participant ethnicity, when the ANOVA already showed no significant difference for coverage error?
Thank you for the feedback.
We revised the paragraph of test result for the participant ethnicity.
There was no statistical difference among the ethnicity groups (ANOVA, p=0.327).
- Figure 3 shows excessive multiple testing without statistical compensation. I recommend dropping figure 3 and include the statistical test values in table 1 for ΔE only.
Thank you for the feedback.
We removed Figure 3 and added statistical test values for ΔE in table 1.
- Figure 5 The color schema for “above threshold” has no match within the graph and column 3 contains a typo (68 % instead of 67 %). Furthermore, figure 5 contains redundant information from table 2. I recommend removing figure 5.
Thank you for the feedback.
We removed Figure 5.
Reviewer 4 Report
The work deals with the spectrophotometric analysis of gingival acrylic shade guides. Five of the commercially available acrylic gingival restorative materials of GC Acrylic, Lucitone 199, IvoCap, Easy-Flow and Candulor. 89 patient patients were contributed to the study with various age, sex and ethnicity. Color measurement was done with the spectrophotometric method and color comparision between shade guides and natural gingiva were calculated in DL, Da, Db and DE. The paper is well structured and procedures are well described, supported by results.
It can be published as it is after adding the bracket after (Figure 1. – Line 83.
Author Response
Comments and Suggestions for Authors
The work deals with the spectrophotometric analysis of gingival acrylic shade guides. Five of the commercially available acrylic gingival restorative materials of GC Acrylic, Lucitone 199, IvoCap, Easy-Flow and Candulor. 89 patient patients were contributed to the study with various age, sex and ethnicity. Color measurement was done with the spectrophotometric method and color comparision between shade guides and natural gingiva were calculated in DL, Da, Db and DE. The paper is well structured and procedures are well described, supported by results.
It can be published as it is after adding the bracket after (Figure 1. – Line 83).
Thank you for the feedback.
We added the bracket.
Round 2
Reviewer 1 Report
ok,all the questions have been settled,I agree to accept this article